# Luminal and Basal Subtypes Across Carcinomas: Molecular Programs Beyond Tissue of Origin

**DOI:** 10.3390/cancers17162720

**Published:** 2025-08-21

**Authors:** Celia Gaona-Romero, María Emilia Domínguez-Recio, Iñaki Comino-Méndez, María Victoria Ortega-Jiménez, Rocío Lavado-Valenzuela, Emilio Alba

**Affiliations:** 1Instituto de Investigación Biomédica de Málaga y Plataforma en Nanomedicina-IBIMA Plataforma BIONAND, 29010 Malaga, Spain; celia.gaona@ibima.eu (C.G.-R.);; 2Laboratorio de Biología Molecular del Cáncer (CIMES-UMA), 29010 Malaga, Spain; 3Unidad Clínica de Oncología Médica, Hospital Universitario Virgen de la Victoria, 29010 Malaga, Spain; 4Anatomía Patológica, Hospital Universitario Virgen de la Victoria, 29010 Malaga, Spain; 5Faculty of Medicine, University of Málaga, 29010 Malaga, Spain; 6Centro de Investigaciones Médico Sanitarias (CIMES, Universidad de Málaga (UMA)), 29010 Malaga, Spain

**Keywords:** luminal subtype, basal subtype, pan-cancer, epithelial tumors, molecular profiling, prognostic biomarkers, precision oncology

## Abstract

Luminal and basal molecular subtypes were first described in breast cancer, where they show distinct biological properties and clinical outcomes. Recent research has revealed that these subtypes are also present in many other epithelial cancers, beyond their tissue of origin. Basal tumors are often linked to aggressive behavior, poorer prognosis, and potential sensitivity to treatments such as DNA-damaging agents or immunotherapy. Luminal tumors usually have more favorable outcomes and may respond to hormone-related therapies. In this review, we summarize studies published between 2010 and 2024 that have applied luminal/basal subtyping across different cancer types and within single tumor types. Our aim is to provide readers with a clear overview of how these molecular programs are defined, where they occur, and why recognizing them could help improve cancer classification, prognosis, and treatment strategies.

## 1. Background

Classically, cancer has been classified according to histological distinctions. Advances in research techniques have enabled the study of the molecular foundations of cancer, allowing the identification of major subtypes within individual cancer types. Understanding the common molecular features of these subtypes has relevant clinical implications. Therapy election and the development of targeted drugs rely on this molecular knowledge. Nonetheless, an alternative approach is emerging, revealing super types that transcend the tissue of origin.

Carcinomas have historically been stratified by their anatomical site of origin and histology. They arise from polarized epithelial tissue, which is organized into distinct apical and basal domains. Epithelial tumors display relative degrees of being luminal or basal that eventually may impact cancer outcomes [1]. Clinically, luminal subtypes are often associated with favorable prognosis and response to hormone-targeted therapies, while basal subtypes tend to show more aggressive behavior, poorer survival, and in some contexts, greater sensitivity to DNA-damaging agents or immunotherapy. These prognostic and therapeutic implications highlight the clinical value of luminal/basal stratification across carcinomas, which has been widely described most extensively in breast cancer [2,3], but also in bladder [4,5], prostate [6], or lung cancer [7,8].

In all carcinomas, epithelial cells can exhibit basal or luminal phenotypes [9,10]. These phenotypes can be distinguished using antibodies against specific keratins (KRT). Keratins are structural proteins that provide mechanical support and external resistance and are routinely used as markers in cancer diagnosis. Luminal cells typically express KRT8 and KRT 18, while basal cells often express KRT5/6 and KRT 14. Keratin expression reflects epithelial cell type, tissue growth, and differentiation status [11,12,13]. The association between keratin phenotype and different clinical and pathological parameters has been extensively investigated. Distinct differentiation statuses have biological and clinical relevance; luminal keratin expression indicates a differentiated phenotype, associated with good prognosis [10,14,15]. Conversely, basal keratin expression correlates with aggressive tumor behavior and poorer patient outcomes [9,14]. Nevertheless, the basal phenotype is not a static entity but rather a dynamic transcriptional state that tumors can acquire over time. During cancer onset, development, and in response to treatment, epithelial cancer cells have the capacity to interconvert between basal and luminal differentiation states [16]. Recent research shows that luminal-to-basal plasticity can occur, leading to intra tumoral lumino-basal heterogeneity and supporting the idea that the basal program is not tissue-dependent, but driven by transcriptional reprogramming [17,18]. This plasticity also plays a crucial role in metastasis onset [9], as epithelial basal cells are more aggressive in forming metastasis than luminal cells. However, the molecular mechanisms underpinning differentiation states and their impact on metastasis remain incompletely understood. Importantly, evidence on the extent and biological relevance of luminal to basal plasticity differs across tumor types, and further studies are needed to clarify how this phenomenon translates into prognosis and therapy in specific clinical contexts.

These observations have been complemented and validated by large-scale cancer genomics initiatives, which provide an unbiased framework to detect and compare luminal and basal molecular features across different tumor types. Since 2008, The Cancer Genome Atlas (TCGA) program has molecularly characterized over 20,000 primary cancers representing 33 cancer types [19]. Over more than a decade, this landmark cancer genomics research network—culminating in the TCGA Pan-Cancer Atlas—has revealed molecular commonalities transcending tumor types, including recurrent driver mutations, pathway alterations, DNA damage detection and repair mechanisms, immune microenvironment features, and stem-ness signatures [20]. Large-scale, multi-omic analyses have classified tumors not only by primary tissue and histology but also into cross-cutting molecular subtypes, some of which correspond to luminal- or basal-like profiles observed across diverse carcinomas [21], reinforcing the concept that these molecular programs are not restricted to a specific tissue. Over the past two decades, accumulating evidence for luminal and basal subtypes across diverse epithelial derived tumors has led to the hypothesis that these subtypes reflect fundamental tumor biology and have important clinical relevance. Breast cancer was the first model of study in which these molecular programs were characterized in detail, revealing clear differences in gene expression profiles, mutational landscapes, and treatment responses [22]. Insights gained from breast cancer research have since guided the search for similar subtypes in other major epithelial cancers, including bladder, head and neck, and lung carcinomas. In these tumor types, luminal and basal features retain many of their original molecular characteristics and clinical associations, supporting the concept that they represent shared biological programs rather than tissue-specific phenomena. This understanding provides the basis for exploring their distribution, prognostic value, and therapeutic implications across carcinomas, which is the focus of the present review.

The key studies that first identified basal-like subtypes in other carcinoma types are summarized in Figure 1 (left panel). Successive early studies reported that several molecular features are common to the breast basal-like, squamous cell lung carcinomas (SQCLCs), and high-grade serous ovarian tumors [3,7,23,24,25]. These shared features across the three cancer types are shown in Figure 2. Additionally, multiple investigations have attempted to classify or characterize large sets of epithelial tumors irrespective of tissue of origin [1,7,21,24,26]. Most relevant papers and their main contributions are represented chronologically in Figure 1 (right panel).

In addition to the individual studies highlighted in this review, large-scale cancer genomics repositories provide an essential framework for contextualizing luminal and basal subtypes across tumor types. The Catalogue Of Somatic Mutations In Cancer (COSMIC) curates detailed data on somatic mutations, copy-number changes, and gene fusions from millions of tumor samples, along with functional annotations such as the Cancer Gene Census [36]. cBioPortal offers an interactive platform integrating TCGA and other datasets for visualization and cross-cancer comparisons [37,38]. Analyses through these platforms have shown that basal-like molecular programs—characterized by TP53 mutations, proliferation signatures, and specific keratin expression patterns—are enriched in multiple carcinomas, further supporting their consistency across tissues. In the following sections, we use breast cancer as the initial model to describe luminal and basal programs in detail, and then extend the analysis to other epithelial carcinomas, highlighting shared molecular traits, clinical implications, and potential therapeutic opportunities.

## 2. Breast Cancer as the First Model of Study

Breast cancer is the most frequent malignancy in women worldwide and is characterized by marked molecular heterogeneity. For more than 20 years, research has taken this heterogeneity into consideration, making breast cancer the first tumor type in which luminal and basal subtypes were identified and their clinical implications extensively investigated. A comprehensive overview of the main concepts and the most widely used classifications of breast cancer subtypes is summarized in Figure 3.

### 2.1. Breast Cancer Histology and Biology

Breast cancer arises in the terminal duct lobular units of the collecting ducts. Cells from normal breast epithelium evolve to invasive breast carcinomas in a complex process involving multiple cellular and molecular alterations [39]. The two most frequent histological subtypes of breast cancer are invasive ductal carcinoma (IDC, also known as invasive carcinoma of no special type) and invasive lobular carcinoma (ILC). IDC is the most common type of breast cancer (70–75%) and starts in the cells lining the milk duct, while ILC is less common (10–14%) and arises from the lobules. Their preinvasive counterparts are referred to as ductal and lobular carcinoma in situ, respectively. There are some other 17 types of breast carcinomas, known as carcinomas of special type (e.g., mucinous, papillary, tubular carcinoma), which account for less than 5% of all breast cancers [40].

Beyond histological distinctions, breast carcinomas display diverse biological features that influence tumor behavior and therapeutic response. Tumor histological grade (G1, G2, or G3) is a well-established prognostic factor, reflecting the degree of differentiation and potential aggressiveness. Immunohistochemical (IHC) assessment of estrogen receptor (ER), progesterone receptor (PR), human epidermal growth factor receptor 2 (HER2), and Ki-67 proliferation index is routinely used in diagnosis, prognosis, and treatment selection. Hormone receptor-positive tumors express ER and/or PR, while those lacking ER, PR, and HER2 expression are classified as triple-negative breast cancer (TNBC) [40,41].

Genetic and epigenetic alterations are involved in breast cancer onset and progression. Frequently altered genes include TP53, PIK3CA, MYC, PTEN, CCND1, ERBB2, FGFR1, and GATA3, which regulate cell cycle control, apoptosis, and oncogenic signaling pathways [40].

The tumor microenvironment also has an impact on breast carcinogenesis and metastasis. The great variability existing in the extracellular matrix, immune cells, and blood cells involved further contributes to the heterogeneity observed in this disease [39,40].

Several classifications on the basis of histological and molecular characteristics have been developed. In breast cancer, the currently established intrinsic molecular subtypes are Luminal A, luminal B, HER2-enriched (HER2-E), basal-like, claudin-low, and normal-like [2,42]. Intrinsic subtypes are shown to be useful in predicting patient survival in breast cancer [43]. Multiple gene expression signatures have been designed for clinical use, including Oncotype DX, MammaPrint, EndoPredict, and PAM50 (Prediction Analysis of Microarray 50), among others. The PAM50 gene signature, widely applied in research and clinical practice, allows breast cancer stratification into four intrinsic subtypes (Luminal A, luminal B, HER2-E, and basal-like) (Figure 3). The algorithm is based on differential gene expression analysis of a minimum set of genes obtained by Parker et al. [28]. This gene signature has demonstrated that the enormous biological diversity of breast cancer observed in DNA, miRNA, and protein studies is captured in the four main intrinsic subtypes defined solely by differential gene expression profile and that, contrary to popular belief, conventional clinicopathological variables are not able to reflect the intrinsic biology of the tumor [28]. Multiple studies have now evaluated and demonstrated the clinical utility of intrinsic subtyping in a variety of clinical settings, and in combination with other biomarkers [44,45,46,47,48]. In clinical practice, surrogate classification is used to guide prognosis and management (Figure 3). Surrogate intrinsic subtypes are based on histology and IHC markers (ER, PR, HER2, and Ki67). While there is substantial concordance between clinical classification and intrinsic molecular subtypes, most basal-like breast tumors fall within the TNBC category. Basal-like breast tumors typically exhibit high-grade, poor differentiation, high proliferative capacity, and TP53 mutations, resulting in an aggressive and relapse-prone phenotype [22,40].

### 2.2. Prognostic Implications in Breast Cancer

Prognostication, therefore, relies on tumor histological and molecular data. Generally, luminal A tumors are featured by good patient prognosis. These tumors present low risk features, such as low-grade, high expression of ER and PR, and low proliferation index. The Luminal B and HER2-enriched subtypes show intermediate prognosis. The Luminal B subtype expresses ER, while PR is less or not expressed, and displays a high-grade and is highly proliferative. The HER2-enriched subtype (HER2^+^ and low or absent expression of ER and PR) is characterized by high-grade and medium-to-high proliferation index. HER2-enriched tumors are aggressive but respond to targeted therapies, resulting in an intermediate prognosis. TNBCs (ER^−^, PR^−^, HER2^−^) have the poorest prognosis. This subtype is prone to early recurrences and is highly aggressive, displaying high-grade and high proliferation index [40], exemplifying both the challenges and recent advances in subtype-based treatments [49].

### 2.3. Management Strategies in Breast Cancer

Treatment strategies in breast cancer are based on the molecular subtype. Locoregional treatment (surgery and radiation therapy) and systemic therapies are the main management options (Figure 3).

In early breast cancer, therapeutic goals include tumor eradication and prevention of recurrence. Non-metastatic breast cancer is considered operable and can be treated with surgery to remove the tumor. Most patients also require some form of systemic therapy, although the specific indication depends on molecular subtype, tumor burden, and absolute risk of recurrence [22,40,50]. Systemic therapies include chemotherapy, endocrine therapy (for hormone receptor-positive tumors), anti-HER2 therapy (for HER2-positive disease), bone-stabilizing agents, poly(ADP-ribose) polymerase (PARP) inhibitors (for cancers with specific DNA-repair defects, including BRCA mutation carriers), and, more recently, immunotherapy [40,51,52,53]. Systemic therapies can be administered before or after surgery (referred to as neoadjuvant or adjuvant therapy, respectively). Neoadjuvant chemotherapy (NAC) is preferred for patients whose tumor burden requires reduction before surgery or when pathological complete response (pCR) information is valuable (as in TNBC and HER2-positive tumors) [54]. Adjuvant chemotherapy can be given if biomarkers or surgical results indicate a high risk of recurrence [40,50].

Endocrine therapy is recommended for all ER-positive and/or PR-positive tumors to inhibit ER activity. The key question for these luminal patients is determining which individuals would benefit from adding chemotherapy in addition to endocrine therapy. In luminal tumors, the decision to administer chemotherapy (NAC or adjuvant) is strongly influenced by tumor proliferation rate and recurrence risk, as assessed by a gene expression signature (GES). Typically, luminal A patients (low proliferation rate, low-risk GES) with low tumor burden can omit chemotherapy [50]. Multiple genomic panels have been clinically validated to guide chemotherapy decisions in breast cancer, and their use is recommended by major clinical guidelines. In addition, clinical assays assessing late recurrence risk in luminal breast cancer may be used to support extended endocrine therapy [40].

Because TNBC lacks ER, PR, and HER2 expression, treatment decisions are more challenging. TNBC is usually treated with chemotherapy (preferably in the neoadjuvant setting) with or without targeted therapy. In HER2-positive breast cancer, NAC is also the standard of care, together with anti-HER2 targeted therapy. Achieving pCR in TNBC and HER2-positive tumors is associated with improved patient outcomes [40,50,54]. The addition of a platinum compound to chemotherapy regimens can increase pCR rates in TNBC, although its benefit on patient outcomes must be weighed against the potential increase in toxicity [40].

The management of advanced breast cancer is also based on subtype, but it differs from the approach for early-stage disease, as locally advanced and metastatic breast tumors cannot be completely removed surgically. Both systemic and locoregional treatments are considered, with the aim of relieving symptoms and prolonging life expectancy [40,50].

Ultimately, breast cancer comprises multiple subtypes with distinct risk profiles and therapeutic approaches. The future direction of breast cancer management is toward individualized treatment, tailoring therapy to the tumor’s molecular subtype.

Importantly, intrinsic subtype classification has already been translated into routine clinical practice through Prosigna© Breast Cancer Prognostic Gene Signature Assay, a PAM50-based gene expression test. Its analytical validity has been demonstrated in multiple laboratories using formalin-fixed breast cancer samples [48]. Prosigna not only assigns tumors to intrinsic subtypes but also provides a risk of recurrence (ROR) score that predicts the probability of recurrence within 10 years. Clinical trials involving over 1000 patients showed that the Prosigna ROR score provides valuable prognostic information in ER^+^ node-negative postmenopausal patients, distinguishing intermediate and high-risk groups [44]. Further studies confirmed that ROR adds prognostic value beyond conventional clinicopathological factors, including long-term prediction of late recurrences [45,46,55,56]. Although PAM50 intrinsic subtyping and the ROR score were initially approved for risk profiling in postmenopausal women, their value in premenopausal patients remained uncertain. Subsequent studies demonstrated that both PAM50 and the ROR score are also applicable in premenopausal women, with some evidence suggesting predictive effects for tamoxifen and chemotherapy benefit [57,58,59,60]. More recent analyses further support the finding that PAM50 subtyping and ROR provide robust long-term prognostic information in premenopausal cohorts as well [61]. Based on this evidence, the American Society of Clinical Oncology (ASCO) has endorsed the use of PAM50 in ER/PR^+^, HER2^−^, node-negative breast cancer to guide adjuvant chemotherapy decisions [47]. It should be noted that Prosigna is not the only genomic test used in clinical practice; other prognostic signatures, such as Oncotype DX, have also been approved for guiding treatment decisions in breast cancer, each within specific patient populations. This example illustrates how luminal/basal molecular classification can directly influence patient management in real-world clinical settings.

Despite its demonstrated clinical utility, the routine implementation of PAM50-based assays such as Prosigna faces several barriers. The test is relatively expensive and requires specialized equipment, which may limit accessibility in public health services or resource-limited settings. Standardization across laboratories is another challenge, since gene expression assays demand strict technical validation to ensure reproducibility of results. Potential solutions include the development of simplified, lower-cost platforms, the integration of PAM50 into widely available next-generation sequencing panels, and efforts by professional societies to establish standardized guidelines for assay performance and interpretation. As these barriers are addressed, the broader use of intrinsic subtype classification could further enhance personalized treatment strategies in breast cancer.

## 3. Pan-Cancer Studies on Luminal and Basal Subtypes

Pan-cancer research initiatives involve the analysis of multiple cancer types, aiming to identify shared patterns and molecular features implicated across different tumors. The purpose of these investigations is to deepen our understanding of the mechanisms driving cancer development and progression, uncover potential therapeutic targets relevant to various cancer types, and ultimately improve clinical management through more effective targeted therapies.

The first pan-carcinoma luminal/basal subtyping study across epithelial tumors, regardless of anatomical site of origin, was published in 2019 by Zhao et al. [1]. Using the PAM50 gene signature, the authors classified carcinomas as luminal A, luminal B, or basal-like, representing the first RNA-based pan-cancer subtype classification. Their comparison of global gene expression patterns revealed that luminal and basal subtypes are consistently present across all tumor types, showing similar expression profiles, independent of tissue origin. Zhao et al. demonstrated that luminal/basal subtyping is useful for predicting clinical outcomes. This pan-carcinoma stratification emphasizes the biological importance of luminal versus basal subtypes and its translational potential in clinical oncology [1]. A graphical summary of luminal and basal markers, irrespective of the tissue of origin, is provided in Figure 4.

Cytokeratins are structural proteins that form a major component of the cytoskeleton in epithelial cells. In general, cytokeratin expression reflects cell phenotype; basal-like tumors show increased expression of KRT5/6 and KRT14, whereas luminal-like tumors exhibit higher levels of KRT8, KRT18, and KRT20 [1,10,14]. Epithelial–mesenchymal transition (EMT) is a pivotal cellular process in which polarized epithelial cells acquire mesenchymal cell characteristics [16]. This process operates under physiological conditions (embryonic development, wound healing) but also under pathological circumstances (fibrosis, cancer). During EMT, alterations involve the disappearance of intercellular adhesions, loss of epithelial polarity, dynamic restructuring of the cytoskeleton, degradation of basement membranes, etc. [18]. EMT-associated features, such as invasiveness and migratory capacity, are more frequently linked to basal subtypes than to luminal A or B tumors [1]. Basal-like tumors are also the most proliferative, closely followed by luminal B tumors.

Distinct mutational profiles are observed across subtypes. Basal tumors display the highest rates of TP53 and RB1 mutations, as well as copy number (CN) loss. Luminal B tumors also show a significant rate of RB1 mutations, only slightly lower than that of basal tumors [1].

Regarding the hormone receptors that govern the response to hormone therapy, luminal breast tumor subtypes highly express ER, and the luminal A subtype has the highest expression of PR, as expected [1]. Analogous to breast cancer, luminal subtypes of ovarian and endometrial cancer show increased ER expression, with PR most abundant in luminal A endometrial tumors. Androgen receptor (AR) expression is typically high in luminal breast tumors but varies by cancer types, being generally lower in basal subtypes. AR is a key marker in prostate cancer, characteristic of luminal tumors, and is associated with hormone responsiveness, which supports the different androgen deprivation therapy (ADT) sensitivities [4]. In bladder cancer, luminal tumors also display higher AR levels compared to basal subtypes [1].

In terms of clinical outcomes, basal-like tumors are associated with poorer survival outcomes compared to luminal A tumors across all cancer types, while luminal B tumors show intermediate survival, consistent with the breast cancer literature [28].

Collectively, these findings demonstrate that luminal- and basal-like stratification across carcinomas is clinically relevant. Identifying transcriptomic, genomic, clinical, and drug sensitivity commonalities across cancer types highlights the biological significance and translational potential of these subtypes.

## 4. Luminal and Basal Subtypes in Other Major Carcinomas

The concept of luminal- and basalness is not restricted to breast cancer and gene expression-based subtyping and can also be applied to identify luminal–basal subsets in carcinomas from other tissues. The left panel in Figure 1 chronologically summarizes the main studies applying PAM50 and other classifiers to other tumor types. A comparative overview of luminal and basal subtypes across the most frequently studied epithelial tumors is summarized in Figure 5.

### 4.1. Prostate Cancer

Both luminal and basal cells can give rise to prostate cancer [62]. In 2017, Zhao et al. [6] highlighted the need for molecular subtyping in this disease to identify subgroups with specific clinical implications. Prostate and breast cancers share several oncogenic pathways and are influenced by hormones. A slightly modified PAM50 algorithm was applied to stratify prostate cancer samples into luminal A, luminal B, and basal-like subtypes, revealing gene expression patterns consistent with intrinsic breast cancer subtypes [28]. These subtypes differ in biomarker expression, prognosis, and treatment response.

Basal-like tumors are enriched in the basal lineage CD49f signature, while luminal markers such as NKX3.1, KRT18, and AR are elevated in luminal-like tumors, supporting the link between subtypes and the established prostate cancer biology [6]. The androgen activity pathway is enriched in luminal subtypes. MYC signaling is the most enriched pathway in luminal tumors, whereas basal tumors are characterized by the upregulation of genes downregulated by KRAS, consistent with MYC and KRAS expression patterns in luminal-like tumors. Luminal A tumors exhibit low expression of proliferation genes. Clinically, patients with luminal B tumors have the poorest prognosis, followed by basal-like tumors, while luminal A tumors show the most favorable outcomes.

The treatment landscape for prostate cancer is rapidly evolving [63]. The association between androgen signaling and molecular subtypes is particularly relevant, as the androgen receptor constitutes the primary target for systemic treatment tumors [6,63]. Androgen deprivation therapy, often combined with radiation therapy (RT), remains the standard first-line treatment. However, remissions induced by these treatments are of variable duration and disease progression often occurs.

Differences in AR activity are linked to treatment response and molecular subtype [63]. AR function varies widely among prostate tumors and correlates more strongly with basal/luminal status than with clinical variables. Different molecular subtypes are associated with distinct AR activity in treatment-naïve prostate tumors [6,63]. Both luminal subtypes show increased AR expression and signaling; however, only luminal B tumors benefit from postoperative ADT [6]. This diversity in AR activity contributes to the variability observed in treatment outcomes, suggesting that AR activity could serve as an initial step in a decision-making algorithm for personalized therapy [63].

Prostate cancer heterogeneity, tumor clonality, and AR plasticity, under selective pressure by ADT, remain significant challenges. Clinical implementation of AR activity assessment requires further validation to induce maximal remission and prolong overall survival with optimal quality of life for patients.

Additional studies confirm the clinical utility of basal/luminal stratification in prostate cancer. The NRG Oncology/RTOG 0521 Phase 3 trial examined high-risk, localized prostate cancer samples and demonstrated that transcriptomic profiling improves prognostication and identifies patients who benefit from adding docetaxel to definitive radiotherapy with androgen suppression [64]. This reinforces the translational relevance of luminal/basal transcriptomic programs across cancer types.

This stratification approach has also been extended to clinically low-risk prostate cancer [65] and metastatic disease [35]. In metastatic castration resistant prostate cancer (mCRPC), luminal A, luminal B, and basal-like subtypes correlate with distinct genomic alterations and phenotypes, and clusters are largely driven by distinct intrinsic cell differentiation states, proliferation rates, and androgen receptor activity [35]. Pathway enrichment differs among subtypes [35]. Notably, mCRPC subtypes exhibit different clinical behavior and differential therapeutic targets. In terms of treatment responsiveness, luminal A mCRPC responds to ADT and docetaxel, whereas basal tumors are resistant in preclinical models [35]. This contrasts with results observed in localized disease, where luminal B tumors showed greater benefit from postoperative ADT [6].

Parallel classification schemes across studies [6,35,65,66] show consistent biological foundations, reinforcing the robustness of prostate cancer subtyping in molecular and clinical contexts [67]. The high concordance across studies suggests that luminal/basal classification enhances prognostic accuracy and treatment decision-making beyond conventional clinical criteria. Integrating this approach could improve therapeutic selection and identify subtype-specific druggable targets in prostate cancer.

### 4.2. Bladder Cancer

Urothelial bladder cancer is a biologically heterogeneous disease with variable clinical outcomes and responses to therapy. High-grade tumors can be further divided into non-muscle-invasive bladder cancer (NMIBC) and muscle-invasive bladder cancer (MIBC) [68]. Low-grade and high-grade tumors show distinct gene expression profiles and molecular subsets can be further identified within each histologic group [69,70].

Sjodahl et al. [15] established the first molecular taxonomy for urothelial carcinoma (UC) based on integrated genomics, defining five major subtypes (“Lund subtypes”). This classification anticipated the future clinical relevance of molecular taxonomy. While the molecular landscape of NMIBC is still being elucidated [15,71], MIBC has been comprehensively characterized.

Parallel efforts from four independent groups identified the molecular subtypes of MIBC [4,30,31,33]. Although each group argued for the existence of a different number of subtypes (between two and four), they showed a high degree of concordance among them. MIBCs can be categorized into basal-like and luminal subtypes based on gene expression, closely resembling the corresponding subtypes in breast cancer. In all the studies, common features were found between basal-like MIBC subgroup and the well-recognized basal-like subtype of breast cancer. Basal bladder tumors, like basal breast cancers, include a claudin-low subtype, while luminal bladder cancers can be subdivided into p53-like and luminal groups. The BASE47 predictor—based on the expression of 47 genes—was validated as an accurate classifier of high-grade UCs into luminal and basal-like subtypes, with demonstrated prognostic value. Comparative analyses of gene expression profiles show that basal-like bladder cancer shares a strong molecular signature with basal-like breast cancer [30,31] as well as with SQCLC and head and neck squamous cell carcinoma (HNSCC) [31].

Basal MIBCs share biomarkers with basal breast cancers, whereas luminal MIBCs are enriched in luminal breast cancer biomarkers [4,31]. Basal-like bladder tumors overexpress epithelial lineage genes such as KRT5, KRT6A, KRT14, and epidermal growth factor receptor (EGFR) [4,31,33], consistent with observations in basal breast cancer [2], SQCLC, and HNSCC [27,72]. They also overexpress multiple EGFR ligands [33], a feature shared with basal breast cancer and HNSCCs [4]. In addition, basal MIBCs express cancer stem-cell markers [31] and mesenchymal features (TWIST1/2, SNAI2, ZEB2, and VIM) [4], as reported in basal-like breast cancer [73]. Other consistently overexpressed genes include epithelial wound healing genes (KRT16, KRT17, PLAU, PLAUR) and keratinocyte differentiation genes, such as those in the epidermal differentiation complex at 1q21 (S100A7, SPRR1B, SPRR3) [33]. In general, basal-like tumors represent a low urothelial differentiation state, while luminal tumors are enriched in markers of urothelial differentiation [30,31,33].

Basal-like MIBCs are also enriched for pathways related to cancer progression, cell survival, and cell movement [30], and are associated with distinct genomic alterations. These include frequent RB1 pathway changes—such as RB1 mutations/deletions, Cyclin D1 (CCND1) amplification, E2F transcription factor 3 (E2F3) amplification, or Cyclin E1 (CCNE1) amplification [30]. Basal-like bladder cancer is enriched in gene sets associated with basal-like breast cancer and tumor-initiating cells. Thus, genes that regulate urothelial development have been found to be coregulated with genes that are fundamental for breast development and tumorigenesis [30]. TP53 mutations are more common in basal-like than in non-basal-like MIBCs [24,33], although some studies report similar frequencies across subtypes [4]. In contrast, luminal tumors are enriched in activating FGFR3 mutations [4]. Basal-like tumors show EGFR pathway activation and EGFR-dependent growth—high rates of EGFR copy number gains, EGFR ligand overexpression, and increased EGFR phosphorylation—making EGFR an attractive therapeutic target [33].

Basal MIBCs also exhibit p63 activation [4]. Transcription factors involved in basal/stem cell biology in normal urothelium (Stat-3, NFκB, Hif-1, and p63) are considerably activated in basal MICBs. In bladder cancer cells, p63 controls MYC expression, which is elevated in basal MIBCs. Conversely, basal Stat-3 and NFκB transcriptional networks are downregulated in luminal MIBCs.

Histologically, luminal bladder cancers often display papillary features [5], suggesting their origin from low-grade papillary tumors that progress to MIBC [74]. Basal MIBCs, in contrast, frequently show squamous and sarcomatoid features, and significantly higher stage or grade [4,5,31,33,68], consistent with the observation that squamous tumors are particularly aggressive, associated with advanced stage, metastatic disease at diagnosis, and high invasiveness and mortality [4,5,33].

As in basal-like breast cancer, EMT and cancer stem cell markers are enriched in basal-like MIBCs [5]. Under physiological conditions, EMT contributes to wound healing, but cancer cells exploit this process to promote invasion and metastasis [5]. EMT is regulated by the balance between Zinc-finger-enhancer-binding (ZEB) transcription factors and miR-200 expression; ZEB1 and ZEB2 promote EMT, whereas miR-200 members induce epithelial differentiation [5]. Basal-like breast cancer cells are more prone to experience EMT than luminal cells [73], partly due to a bivalent chromatin configuration at the ZEB1 promoter, which supports plasticity and tumorigenicity. Alterations in TP63 pathway and miR-200 expression are hypothesized to be central to EMT regulation [73,74].

Recent studies have shown that lentiviral knockdown of SOX2—a known transcription factor and biomarker of urothelial cancer stem cells—reduces the expression of stem-associated proteins, oncoproteins, and basal keratins [75]. SOX2 inhibition induces luminal marker expression and improves chemotherapy response, supporting a model in which basal phenotype maintenance is driven by specific transcriptional programs that can be therapeutically targeted.

The tumor microenvironment may influence tumor identity, but its impact in bladder cancer remains uncertain. A longitudinal multiomics analysis using patient-derived xenograft models across different anatomical sites found that bladder cancer subtypes exhibit limited plasticity across different microenvironments and in metastases since no emerging intratumor heterogeneity or subtype transitions were observed [76]. These tumors maintained their molecular subtype, as well as their transcriptomic and genomic profiles, across different environments, demonstrating that bladder cancers have a strong identity that is not easily reprogrammed by the environment. While basal-like features may arise through transcriptional plasticity in some tumors, evidence from bladder cancer suggests that molecular subtypes may remain stable in different microenvironmental contexts. This finding highlights that the extent of luminal-to-basal plasticity may vary across tumor types.

Clinically, MIBC subtypes are meaningful; basal-like tumors are associated with poor prognosis, showing significantly reduced disease-specific and overall survival [4,30,33]. They also respond differently to neoadjuvant chemotherapy, like their breast cancer counterparts. Basal tumors derive greater benefit from NAC and have better long-term outcomes than luminal tumors [5]. EGFR has been proposed as a therapeutic target in basal MIBCs [4,33,77], and preclinical models confirm sensitivity of basal-like MIBC cells to anti-EGFR therapy [33].

Finally, machine learning models have been developed to predict basal and luminal bladder cancer subtypes [78]. This approach, trained on routine pathology slides, suggests that the integration of artificial intelligence tools into the diagnostic process could improve the accuracy of molecular subtyping or enable molecular classification when transcriptomic data is unavailable. Furthermore, this tool supports the feasibility of non-invasive detection of basal features across tumor types.

In summary, MIBC subtypes are highly reminiscent of those in breast cancer. Their molecular characterization holds important implications for prognostication, targeted therapy development, and clinical management. The feasibility of assigning intrinsic subtypes in bladder cancer using only 47 genes [30] supports the potential integration of basal/luminal subtyping into routine clinical practice.

### 4.3. Lung Cancer

There are notable similarities between breast and lung carcinomas, particularly regarding the role of estrogen in driving tumor progression [79]. The PAM50 gene panel has proven to be prognostic in both adenocarcinoma and squamous cell lung carcinoma, representing the most frequent histologic subtypes in non-small cell lung cancer (NSCLC). Genes within the PAM50 signature are of prognostic value in lung cancer, especially SQCLC [79], consistent with findings from pan-cancer analyses which highlighting strong similarities between basal-like breast cancer and SQCLC [7,23,24,80].

Both tumor types share common expression patterns characterized by elevated expression of genes normally enriched in basal epithelial cells [23,24]. They also exhibit high rates of TP53 mutation, large-scale copy-number changes, and increased expression of immune and proliferation pathway genes, leading to high proliferation rates and similar immune profiles [23,24,80].

From a clinical perspective, genes in the PAM50 panel, particularly those linking ER and HER2 signaling, have been shown to provide a predictive tool in lung cancer patients [79].

Other attempts to cluster lung tumors have found that classification into expression subtypes recapitulates histologic subtypes, grouping tumors into squamous, large-cell, small-cell, and adenocarcinoma categories [81]. This supports the hypothesis that each histologic subtype may originate from a distinct progenitor cell: in particular, that the progenitor of SQCLC would have basal epithelial features.

Alternative clustering method developed by Wilkerson et al. [8] determined that the genetic and clinical heterogeneity of SQCLC can be stratified in four mRNA expression subtypes: primitive, classical, secretory, and basal. These subtypes differ in their clinical outcomes and show expression patterns corresponding to distinct normal lung cell types. Interestingly, the SQCLC basal subtype exhibits similarities to normal lung basal cells [8], basal-like breast cancer, and a head and neck squamous cell carcinoma subtype (‘Group 1’ by Chung et al. [27]).

As in breast cancer, it has been proposed that the distinct SQCLCs subtypes arise from different progenitor cells, which may explain the observed differences in mRNA expression and their resemblance to normal lung cell populations [8]. The expression profiles of these subtypes are linked to unique biological processes, arguing for targeted interventions exploiting these differences [8].

If successfully integrated into clinical practice, applying these molecular subtypes could enable more precise prognostication and personalized management of lung cancer patients.

### 4.4. Head and Neck Cancer

Head and neck cancer is a heterogeneous disease whose prognosis is mainly based on location, tumor size, and the presence of lymph node metastases or distant metastases [27,82,83]. Gene expression profiling has proven valuable for stratifying head and neck squamous cell carcinomas (the most common histology) into subtypes in several studies, offering a promising avenue for identifying clinically relevant biomarkers [27,29,34].

Independent gene expression-based studies have consistently defined four molecular classes of HNSCC, showing a high degree of concordance and similar expression patterns across studies [27,29,83]. An alternative taxonomy identifies five distinct HNSCC subtypes, accounting for human papillomavirus (HPV) status [34]. The strong correlation to the previously identified subtypes [27] reinforces the validity of this classification.

Expression patterns in HNSCCs resemble those identified in breast or lung carcinomas and are consistent with SQCLC subtypes, further supporting the idea that tumors from different tissues may share common biological pathways for tumor development and metastasis [8,27,29,34]. The basal subtype of HNSCC shares marked expression similarities with basal-like breast cancer [2] and lung squamous carcinomas [81], and is associated with the poorest clinical outcomes [27].

Basal HNSCC and basal-like breast tumors both overexpress Bullous Pemphigoid Antigen 1, P-Cadherin (CDH3), Laminin γ 2 (LAMC2), and Collagen XVII-α (COL17A1) [2,42]. Likewise, basal HNSCCs and lung squamous tumors share expression of Bullous Pemphigoid Antigen 1, COL17A1, FGF-BP, and Kallikrein 10 [27,29,81]. These patterns are associated with poor patient outcomes both in lung and breast cancers. Basal HNSCCs are also characterized by an active hypoxia signaling and EGFR pathway activation [27,34], suggesting potential benefit from EGFR inhibitor therapies.

HNSCC subtypes display distinct patterns of chromosomal gain and losses in key regions [29,34] that are eventually reflected in expression profiles. Notably, the basal subtype consistently overexpresses TP63 and ITGB4 [29,34].

In HNSCC, metastasis risk can be predicted using expression patterns that are involved in breast metastasis prediction [23,27]. The shared biological behavior and expression features across these tumor types point to common mechanisms driving epithelial tumor metastasis.

### 4.5. Pancreatic Cancer

Pancreatic ductal adenocarcinoma (PDAC) is the most common form of pancreatic cancer (>90%) and remains the deadliest cancer worldwide. Prevention and early detection are currently ineffective in this disease, resulting in 80% of patients displaying incurable stages at diagnosis. Pancreatic tumors are very prone to local and distant metastases after surgery, and chemotherapeutic approaches are unsuccessful [84,85,86].

PDAC is a heterogeneous disease encompassing multiple histologic variants [87]. Histologically, this disease can be diagnosed by its distinctive pathological characteristics; neoplastic cells from the exocrine pancreas invade the surrounding stroma inducing an inflammatory and desmoplastic reaction [86,88].

From a biological perspective, PDAC is an extremely complex malignancy. It is characterized by extensive stromal involvement and a high degree of immune evasion, both of which contribute to therapy resistance [84,89]. Four major driver genes have been identified: KRAS, CDKN2A, TP53, and SMAD4 (one oncogene and three tumor suppressor genes, respectively) [88]. Beyond these, genetic susceptibility to sporadic forms of the disease has been linked to up to 40 common genetic variants, which together may explain a significant portion of disease risk [84]. Recurrently mutated genes can be grouped into 10 key pathways: KRAS, TGF-β, WNT, NOTCH, ROBO/SLIT signaling, G1/S transition, SWI-SNF, chromatin modification, DNA repair, and RNA processing [90].

Transcriptomic analyses on PDAC biology have defined and validated two main gene expression subtypes for the tumor tissue (‘classical’ and ‘basal-like’), with different molecular profiles and clinical outcomes [84,87]. The classical subtype is characterized by the expression of pancreatic progenitor markers and regulation by GATA6 (standing out as a potential biomarker for classical subtype tumors). Basal-like PDAC tumors (also referred to as squamous/quasi-mesenchymal by some authors) display transcriptomic profiles similar to those of basal-like bladder and breast tumors, including EMT characteristics and TP63 expression [85,87,89,90]. Recently, MED12 has been identified as a key regulator of the basal cell state in PDAC, working in cooperation with the transcription factor ΔNp63 to activate basal phenotype transcriptional programs, reinforcing the aggressive nature of this subtype [91]. This highlights potential opportunities for tumor lineage-directed therapies. Such evidence supports the concept of molecularly regulated basal phenotypes in non-mammary epithelial tumors, suggesting shared pan-cancer mechanisms.

Histologic differentiation grade also varies; classical tumors are often well differentiated, while basal-like tumors tend to be poorly differentiated. Subtypes for the stromal compartment have also been proposed (‘normal’ and ‘activated’) [84,87,89]. PDAC exhibits marked spatial phenotypic heterogeneity, associated with differences in immune infiltration and therapeutic response [92,93]. Notably, classical and basal-like programs can coexist in this disease. Basal-like features influence the immune landscape and have implications on treatment strategies.

Across all studies, the basal-like subtype is consistently associated with poorer survival compared with the classical subtype [84,87]. In metastatic disease or locally advanced PDAC, systemic chemotherapy remains the mainstay for prolonging progression-free and overall survival (OS), while for the remaining minority with localized resectable malignancies (20% of PDAC), surgery is the only potentially curative approach [84]. The addition of potent adjuvant and neoadjuvant chemotherapy regimens have achieved some improvements in patient outcomes. However, basal-like PDAC typically responds poorly to chemotherapy, and, to date, no effective subtype-specific treatment strategies have been validated. Many questions remain to be addressed concerning optimized treatments for this cancer type.

Overall, although transcriptomic analyses consistently support the existence of classical and basal-like PDAC subtypes with distinct prognoses, the translation of this knowledge into clinical practice is still emerging. The utility of subtype classification for guiding therapeutic decisions remains limited, as no validated subtype-specific strategies are available. Moreover, variability in basal features across patient cohorts and the coexistence of classical and basal programs within the same tumor add complexity. These uncertainties highlight the need for further studies before luminal/basal classification can be routinely applied in PDAC management. Moving forward, research in this malignancy represents both a challenge and an opportunity to deepen our understanding of the molecular determinants of this disease, with the ultimate goal of developing more effective prevention, early detection, and therapeutic approaches. A promising step will be the design of a practical classifier suitable for routine diagnostics.

## 5. Conclusions and Outlook

### 5.1. Basal Hallmarks

Biological processes and signaling pathways are not restricted to a single tissue or anatomical site. Instead, multiple molecular mechanisms operate simultaneously across large subsets of human tumors originating from different tissues. The exploration of basal features across diverse tumor types has revealed a striking convergence in fundamental cellular and molecular characteristics associated with tumorigenesis beyond the cancer site of origin.

The consistent identification of basal-like phenotypes underscores the universality of basalness across multiple cancer types. The basal-like subtype is defined by a distinctive mutational profile, a unique gene expression signature, and a characteristic clinical behavior, all of which are independent of tumor location. In basal carcinomas, alterations in key signaling pathways give rise to tumors characterized by a highly proliferative and poorly differentiated phenotype. EMT-like features are also a hallmark of basal-like tumors, conferring increased migratory and invasive capacity. Furthermore, basalness is associated with resistance to apoptosis. Together, these traits lead to aggressive tumors with poor patient prognosis.

Overall, the recurrent commonalities observed across carcinomas in transcriptome, genome, clinical outcomes, and drug sensitivity, underscore the biological and translational significance of luminal versus basal subtypes.

### 5.2. Putative Clinical Implications: Unveiling the Prognostic and Predictive Significance of the Basal Subtype in Epithelial Tumors

The evidence compiled here emphasizes the prognostic implications of the basal subtype in epithelial tumors, revealing a shared molecular landscape that extends beyond well-established basal-like breast cancer. Numerous studies across diverse malignancies have consistently linked the basal subtype to aggressive clinical behavior, poorer outcomes, and increased metastatic potential. Beyond its prognostic significance, the basal subtype has emerged as a potential predictive biomarker with important implications for treatment selection. Collectively, these findings suggest that patients with luminal and basal tumors should be considered for distinct therapeutic approaches. The unique molecular features of the basal subtype may influence responses to specific therapies, opening the door to more precise and targeted interventions. Stratifying patients by luminal/basal molecular diagnosis could enable clinicians to tailor therapeutic regimens, ultimately improving treatment efficacy and patient outcomes.

The identification of common molecular signatures underlying basalness provides new opportunities to develop targeted interventions and personalized therapeutic strategies that cut across traditional histological boundaries. Designing therapeutic approaches based on these shared molecular features, rather than on tissue of origin distinctions, represents both a challenging path and a promising avenue. Clinically available agents such as pembrolizumab [94], anti-HER2 therapy [95], or drugs targeting TRK fusions [96], provide encouraging evidence supporting this approach.

However, a key limitation of personalized medicine lies in the feasibility of comprehensively assessing each patient to determine the specific altered pathways driving tumor behavior and subsequently selecting the most appropriate targeted therapy. While luminal/basal subtyping was originally developed for breast cancer, it has been successfully applied to non-breast epithelial tumors, though terminology and specific gene signatures may differ across tumor types. The next major challenge will be to develop a pragmatic classifier that can be integrated into routine diagnostics to identify intrinsic molecular subtypes in a wide range of non-breast epithelial tumors. Such taxonomy will require rigorous clinical validation in successive studies. The aim should be to systematically characterize luminal and basal subtypes in non-breast epithelial malignancies, correlating them with clinical outcomes and response to treatment.

Understanding molecular subtypes, which are intrinsically linked to cell of origin, genetic alterations, and pathway dysregulation, could pave the way toward a unified classification system and enable subtype-specific therapeutic targeting. Incorporating a luminal/basal molecular taxonomy in clinical practice would significantly improve the accuracy of prognostic predictors and optimize therapeutic choice.

As we continue to unravel the complexities of basalness across tumor types, this emerging paradigm has the potential to transform our understanding and treatment of cancer, advancing precision medicine and offering new avenues for effective, tailored interventions, regardless of the cancer site of origin.

## Figures and Tables

**Figure 1 cancers-17-02720-f001:**
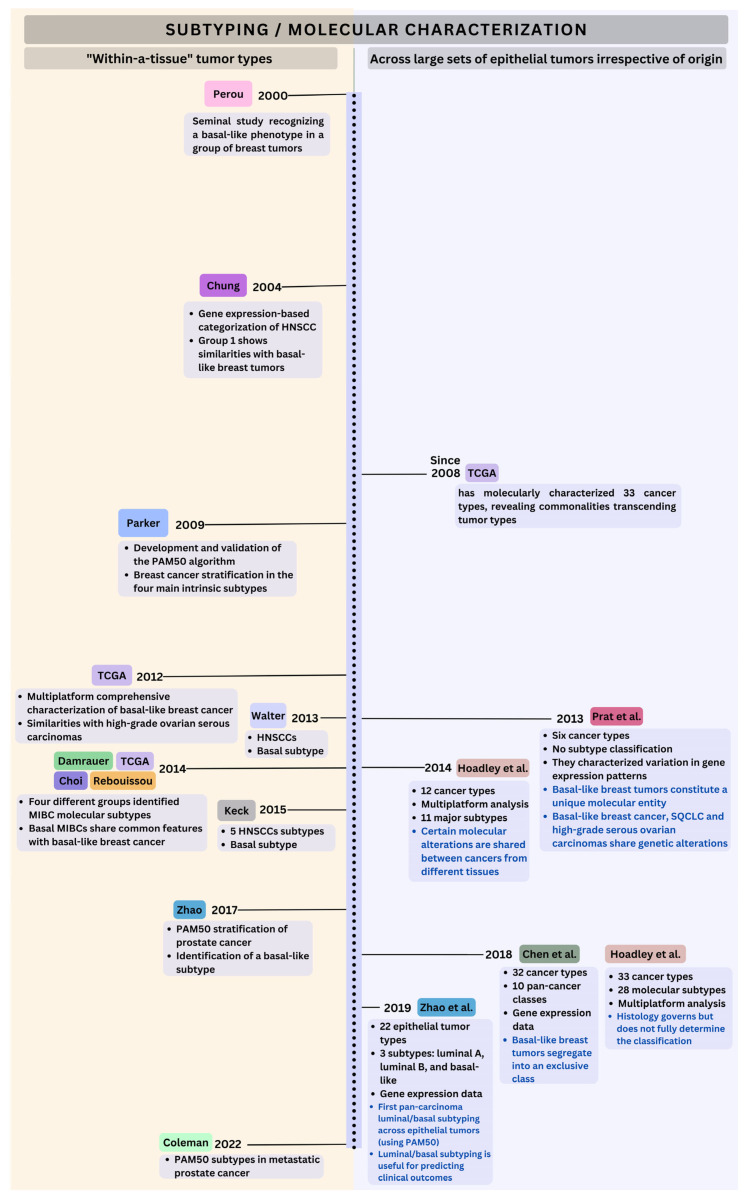
Timeline of advances in molecular characterization of basal and luminal subtypes. Research works performing luminal/basal subtyping or molecular characterization are represented chronologically. Studies analyzing large cohorts of carcinomas irrespective of tissue of origin are shown in the right panel [1,7,21,24,26], while studies focusing on individual tumor types are shown in the left panel [2,3,6,27,28,29,30,31,32,33,34,35].

**Figure 2 cancers-17-02720-f002:**
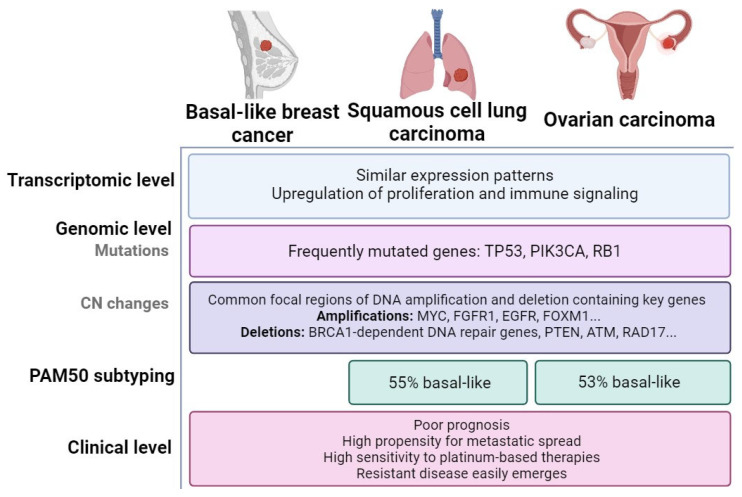
Common features of basal-like breast, SQCL, and ovarian carcinomas. Transcriptomic analyses show that basal-like breast tumors are more similar to SQCLCs than to high-grade serous ovarian carcinomas and the rest of breast tumors. These three tumor types also share alterations at the genomic level, including mutations and large-scale copy-number changes involving key genes. Application of PAM50 intrinsic subtyping to non-breast tumors identified the basal-like subtype in 55% of SQCLCs and 53% of ovarian cancers.

**Figure 3 cancers-17-02720-f003:**
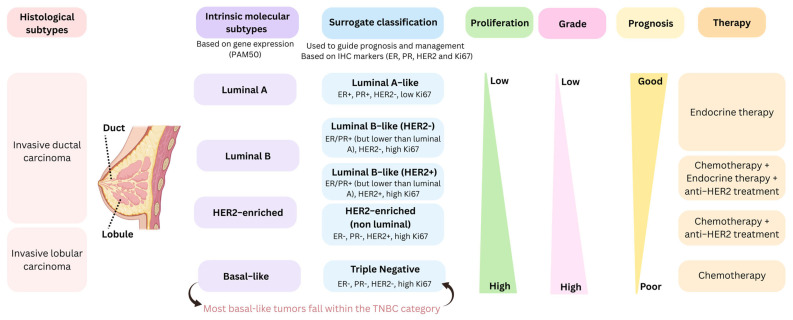
Breast cancer subtypes. Histological subtypes (ductal and lobular) can be further stratified into intrinsic molecular subtypes defined by PAM50 gene expression profiling and surrogate immunohistochemistry-based classification. These categories differ in proliferation, grade, prognosis, and recommended therapeutic strategies.

**Figure 4 cancers-17-02720-f004:**
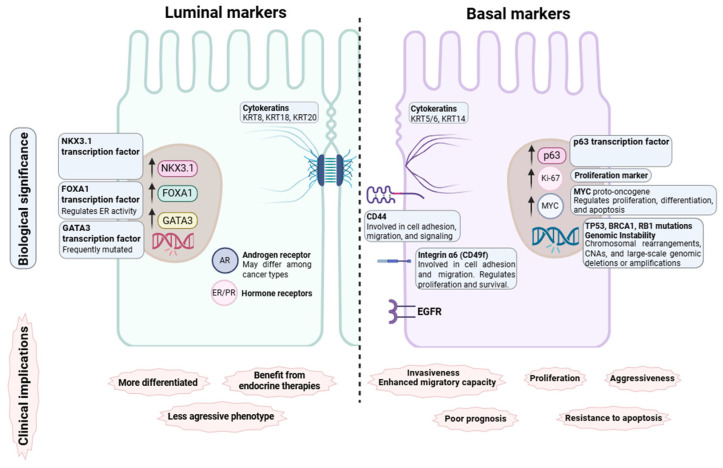
Main luminal and basal markers, biological significance, and clinical implications. The features described can vary within individual tumors and across different studies. Luminal and basal markers can be used to distinguish luminal and basal-like subtypes in cancer diagnosis. These hallmark features are also helpful in cancer management and treatment election since they are associated with tumor behavior and prognosis. Cytokeratins, routinely used to distinguish subtypes, contribute to the cytoskeletal structure of luminal and basal cells. KRT5 and KRT4 are associated with aggressive tumor behavior and poor prognosis, while KRT8 and KRT18 maintain the integrity of luminal epithelial cells and contribute to a less aggressive phenotype. Transcription factor p63 is highly expressed in basal-like tumors, supporting the high proliferation rates that often characterize this subtype. Cell surface proteins, such as CD44 and CD49f, play a role in cell adhesion and underlie migratory capacity and invasiveness related to a more aggressive phenotype. High expression of Ki-67 proliferation marker and EGFR likewise associate with aggressiveness and poor prognosis. On the other hand, hormone receptors are usually associated with a more differentiated and less aggressive phenotype. In addition, ER, PR, and AR status are used to guide treatment decisions. Upregulation of NKX3.1, FOXA1, and GATA3 transcription factors is associated with luminal phenotype. Upregulation of MYC proto-oncogene is associated with basal subtype, which is characterized by genomic instability, contributing to tumor heterogeneity and aggressive behavior. Together, differences between molecular subtypes in gene expression profiles, hormone receptors, mutational profiles, and CN alteration patterns may explain the differing sensitivities to specific therapies and the different clinical outcomes observed between luminal and basal subtypes.

**Figure 5 cancers-17-02720-f005:**
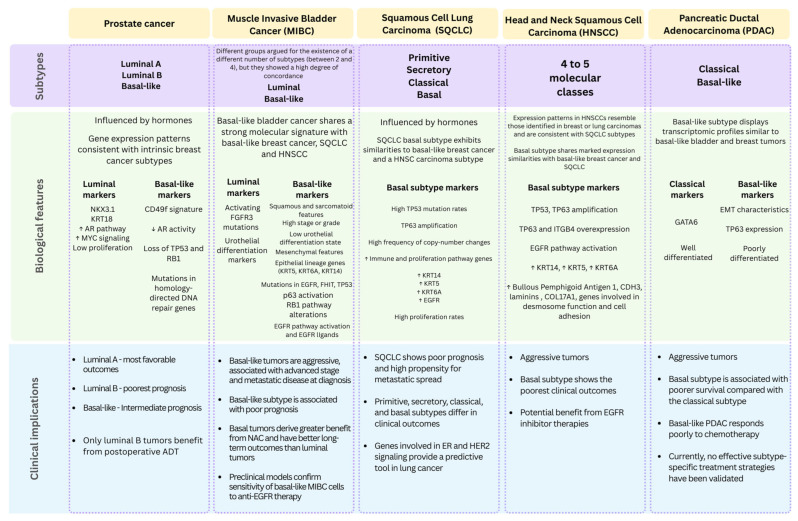
Luminal and basal subtypes across major epithelial carcinomas beyond breast cancer. The scheme summarizes biological features, subtype-specific markers, and clinical implications in prostate cancer, muscle-invasive bladder cancer (MIBC), squamous cell lung carcinoma (SQCLC), head and neck squamous cell carcinoma (HNSCC), and pancreatic ductal adenocarcinoma (PDAC). Across these tumor types, basal-like subtypes consistently associate with aggressive biology, poorer prognosis, and distinct therapeutic vulnerabilities. ↑: Increased expression; ↓: Decreased expression.

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
