# Peer review of "Luminal and Basal Subtypes Across Carcinomas: Molecular Programs Beyond Tissue of Origin"

_cancers, 2025, doi:10.3390/cancers17162720_

Round 1

Reviewer 1 Report

Comments and Suggestions for Authors

In this review, the authors describe several studies that take into consideration connections between cancers that go beyond histology. They make interesting arguments based on existing studies that molecular markers exist that can be used to connect different cancers if one does not restrict their analysis on tissue of origin. This is indeed true.

My major concern is that the review is too superficial and does not take into consideration the entire available data. The authors focus on several specific studies to make their point but ignore much larger studies used by most investigators.  For example, TCGA is barely mentioned. Other repositories such as COSMIC and cBioPortal are not even addressed. These repositories classify data from each sample in much more detail than just mutation type or gene expression. For example, TCGA data is classified by primary tissue as well as primary histology (e.g. carcinoma, glioma, etc) and secondary histologies (basal cell carcinoma, ductal carcinoma, etc). These subclasifications are linked with mutation type and other forms of molecular data. One would think that it would be possible to use data from these repositories to make these connections among cancers. I, therefore, recommend that he authors dig deeper into available literature and describe these other more major studies. As it stands now, it seems that the authors picked and chose only studies that support their argument.

Reviewer 2 Report

Comments and Suggestions for Authors

The review entitled “Basal-ness beyond cancer site-of-origin” is pretty interesting but need to revise major things. After that can be considered too publishable in MDPI “Cancers” journal. The following comments should be addressed.

Comments-1: The title of the review paper must be unique, there is gap between main contain of paper and must be cover broad aspect of the article. Also take a special attention on AI guided notation such as “Basal-ness” and “site-of-origin”. Simple summary and abstract far from the main contain of the article. Both should be one platform to understandable for the reader of basic concept. Some sentence are to long with grammatically error “Moving towards a new paradigm in oncology is ……….individual cancer types rather than on histological distinctions”, check it with carefully and make the sentence into two to three parts. In the abstract author must be provide collection years of the paper such as last five years or last ten years or etc... Main aim of the review article must be elaborately discusses on the respective context to get better idea of the topics.  Keyword is more specific, it should be more border ranges. Graphical representation must be clearer and proper meaning which help to the reader extract idea of the article.

Comment-2: Background of the article are not cover most of the important point of the topic, it should be include in the main text. Also connectivity of two paragraph such as Page no: 3 “line 43-49” and “line 50-55” has lack of flow, so difficult to catch the proper point. In the context some of words such as “tissue-dependent”, “super-types”, “status-specific”, “within-a-tissue “and “epithelial-derived” can be replace with “tissue dependent”, “super types”, “status specific”, “within a tissue” and “epithelial derived” etc to avoid AI generated grammatical correction. In the background section in last paragraph must be minimal contain of next discussion tropic such as “Breast cancer as the first model of study” , etc.. to easier to the reader to get some idea why next paragraph is important for the article.

Comment-3: In the article author represent total three figures. So some of contain must be discussed with pictorial representation such as section “2. Breast cancer as the first model of study” and “4. Most frequently studied tumors (non-breast epithelial tumors)”so reader can easily catch the proper point of the discussion section. The subtitle of the topic “3. Pan-cancer studies by Zhao” must not be a specified with author. The subtitle “4. Most frequently studied tumors (non-breast epithelial tumors)” must not be like “(non-breast epithelial tumors)”.

Comment-4: Overall the article are well organized, vest number of information with appropriate arrangement but many current reference are missing and it should be revised thoroughly.

Reviewer 3 Report

Comments and Suggestions for Authors

This review discusses how the basal phenotype—a set of molecular and functional traits linked to aggressive tumor behavior—appears in many epithelial cancers beyond their tissue of origin. Traditionally, cancers are classified by site and histology, but luminal/basal subtyping reveals shared biology across breast, bladder, lung, prostate, pancreatic, and head & neck cancers. Moving toward biology-driven rather than tissue-based cancer classification could enable more precise, subtype-specific treatments across multiple cancer types

Suggestions for improvement in the review:

Some molecular biology descriptions (e.g., keratin types, EMT pathways) could be streamlined to improve readability for non-specialist audiences without losing key meaning.

Present the prognostic and therapeutic implications of luminal/basal subtyping at the start to engage clinicians before diving into detailed molecular data.

A visual comparing basal vs. luminal traits (across gene expression, prognosis, therapy response) for all discussed cancers could make cross-cancer similarities clearer.

Show how basal/luminal classification has changed management decisions in real-world cases to illustrate impact.

Expand on barriers to routine use of PAM50 or similar classifiers, such as cost, accessibility, and standardization, and suggest possible solutions.

Explicitly note where evidence is still emerging (e.g., pancreatic cancer therapy targeting by subtype, variability in basal plasticity between tissues).

Round 2

Reviewer 1 Report

Comments and Suggestions for Authors

The authors have been responsive to my critiques and made significant changes to this second version. This reviewer is satisfied.

Reviewer 2 Report

Comments and Suggestions for Authors

I have thoroughly reviewed the revised version and found that all comments have been addressed effectively. The revised format is significantly improved compared to the previous version. Therefore, I recommend the revised manuscript for acceptance for publication.